# Biosensor-Integrated Tibial Components in Total Knee Arthroplasty: A Narrative Review of Innovations, Challenges, and Translational Frontiers

**DOI:** 10.3390/bioengineering12090988

**Published:** 2025-09-17

**Authors:** Ahmed Nadeem-Tariq, Christopher J. Fang, Jeffrey Lucas Hii, Karen Nelson

**Affiliations:** Kirk Kerkorian School of Medicine, University of Nevada Las Vegas, 625 Shadow Lane, Las Vegas, NV 89106, USA; christopher.fang@unlv.edu (C.J.F.);

**Keywords:** total knee arthroplasty, electrochemical biosensors, smart implants, implantable sensors, biomedical engineering, orthopedic innovation, personalized medicine, sensor-enabled arthroplasty, translational research, interdisciplinary collaboration

## Abstract

Background: The incorporation of biosensors into orthopedic implants, particularly tibial components in total knee arthroplasty (TKA), marks a new era in personalized joint replacement. These smart systems aim to provide real-time physiological and mechanical data, enabling dynamic postoperative monitoring and enhanced surgical precision. Objective: This narrative review synthesizes the current landscape of electrochemical biosensor-embedded tibial implants in TKA, exploring technical mechanisms, clinical applications, challenges, and future directions for translation into clinical practice. Methods: A comprehensive literature review was conducted across PubMed and Google Scholar. Articles were thematically categorized into technology design, integration strategies, preclinical and clinical evidence, regulatory frameworks, ethical considerations, and strategic recommendations. Findings were synthesized narratively and organized to support forward-looking system design. Results: Smart tibial implants have demonstrated feasibility in both bench and early clinical settings. Key advances include pressure-sensing intraoperative tools, inertial measurement units for remote gait tracking, and chemical biosensors for infection surveillance. However, the field remains limited by biological encapsulation, signal degradation, regulatory uncertainty, and data privacy challenges. Interdisciplinary design, standardized testing, translational funding, and ethical oversight are essential to scaling these innovations. Conclusions: Biosensor-enabled tibial components represent a promising convergence of orthopedics, electronics, and data science. By addressing the technological, biological, regulatory, and ethical gaps outlined herein, this field can transition from prototype to widespread clinical reality—offering new precision in arthroplasty care.

## 1. Introduction

### 1.1. Clinical Context: The Growing Demand and Risk Landscape of TKA

Total knee arthroplasty (TKA) is one of the most common and successful orthopedic procedures globally, with over 1.89 million TKAs captured in the 2024 AJRR report alone [1]. Driven by an aging population, improved surgical access, and expanding indications, TKA volume has grown exponentially across the last two decades. Lewis et al. observed a 384% increase in primary knee replacements in the U.S. and over 127% in Australia from 2003 to 2017 [2]. While revision rates remain relatively low—2.6% at 10 years for cemented TKAs in patients ≥65—the absolute number of revision procedures is steadily rising, straining health systems worldwide.

The burden of revision surgery is particularly pronounced in younger patients (<65 years), who now account for a significant share of revisions due to their longer life expectancy and higher physical demands. Nearly 50% of revisions in the U.S. occur in this demographic, with costs exceeding USD 49,000 per case and a projected national burden of USD 13 billion by 2030 [3]. Furthermore, global disparities in TKA cost-effectiveness are evident—ranging from USD 3457 in India to over USD 19,000 in the U.S.—with existing economic models failing to account for real-time monitoring technologies [4].

While TKA is highly successful overall, failures still occur due to aseptic loosening, periprosthetic joint infection (PJI), instability, and malalignment. Registry data from 11 countries show that revision causes vary regionally—infection dominates in Sweden and Canada, whereas aseptic loosening prevails in Japan and Australia [5]. Approximately one-third of early revision TKAs could be avoidable, often linked to instability and improper fixation, indicating clear targets for early detection [6]. Infection remains a major concern, with deep infection rates of 9.2% among revision cases and comorbidities and index infection serving as strong predictors of recurrence [7].

### 1.2. Rationale for Smart Monitoring: Bridging the Detection Gap

Current surveillance of prosthetic joint performance relies heavily on intermittent clinical evaluation, radiographs, and subjective patient-reported outcomes. However, these tools often fail to capture early, subclinical changes that precede mechanical loosening or infection. Implantable biosensors have been proposed to enable real-time, in situ monitoring in orthopedics, allowing feedback-driven rehabilitation and early risk stratification [8].

Recent case evidence illustrates the potential: embedded inertial sensors in a Persona IQ^®^ implant detected a decline in gait performance before symptom escalation, enabling timely manipulation under anesthesia [9]. Digital biomarkers derived directly from the implant can outperform PROMs and standard follow-up exams in identifying early dysfunction. Similarly, conventional balancing methods are inconsistent under complex loading, supporting the need for quantifiable sensor feedback [10].

Furthermore, imaging often fails to detect early implant degradation, whereas bio-electrochemical sensing can capture corrosion, wear, and micro-environmental changes well before structural or symptomatic failure [11].

### 1.3. Precedent Smart Implants: Intraoperative and Postoperative Innovations

Smart arthroplasty began with intraoperative load sensors such as VERASENSE™, which quantify medial-lateral pressure differentials during TKA to optimize soft-tissue balancing. Achieving balance within 15 lb across compartments has been linked to significantly higher Knee Society Scores (KSS) at follow-up [12]. A meta-analysis of 2147 TKAs found that sensor-guided balancing significantly reduced MUA rates, though PROMs and reoperation rates remained similar to manual balancing [13]. However, long-term functional differences remain modest, highlighting the need for more advanced data beyond static load distribution [14].

More recently, commercial smart implants like Persona IQ^®^ integrate inertial measurement units (IMUs) into the tibial stem, enabling daily remote tracking of gait parameters. This technology has evolved from early prototypes (e-Knee) to modern wireless systems, though key challenges remain in powering, data shielding, and implantation logistics [15]. Current systems focus largely on biomechanical sensing, leaving a gap in biochemical monitoring for infection, inflammation, or implant-material degradation.

### 1.4. Why Electrochemical Biosensors? A Logical Evolution

Unlike inertial or piezoelectric sensors, electrochemical biosensors can monitor local biochemical environments, including pH (infection), lactate (inflammation), nitric oxide (ischemia), and metal ions (degradation). These sensors are miniaturizable, low-power, and capable of broad analyte detection [16].

Electrochemical signatures of corrosion can be captured through embedded sensors, enabling real-time detection of implant degradation [11]. When integrated with data analytics and secure telemetry, these biosensors could power closed-loop systems for infection detection, personalized loading adjustments, or predictive analytics in postoperative care.

### 1.5. Objectives and Outline of This Review

Given the unmet need for real-time, in situ monitoring of prosthetic joint environments and the growing interest in sensorized implants, this narrative review aims to:Synthesize the landscape of implantable electrochemical biosensors relevant to total knee arthroplasty.Evaluate the engineering and integration challenges of embedding these sensors into tibial components.Assess the preclinical and clinical evidence supporting their use.Propose a translational roadmap for clinical deployment.

The review proceeds by analyzing current sensor modalities, the engineering considerations of implant integration, available evidence from animal and human studies, challenges to adoption, and future directions—including AI integration, standardization, and regulatory pathways.

## 2. Literature Selection

This review was developed through a narrative approach aimed at exploring the intersection of electrochemical biosensors and total knee arthroplasty (TKA), with particular emphasis on the integration of sensing technologies into tibial components. Recognizing the novelty and interdisciplinary nature of this field, the selection of literature was guided not by a formal systematic protocol but by thematic relevance, empirical depth, and translational value across engineering, biomaterials, and orthopedic sciences.

The literature search was performed manually using PubMed and Google Scholar as the primary databases. Articles were identified through combinations of terms such as “total knee arthroplasty,” “electrochemical biosensor,” “smart implant,” “tibial insert,” and “embedded sensor.” Additional articles were retrieved by reference mining from key review papers and technical reports, allowing the inclusion of foundational sources that may not have appeared through keyword-based indexing alone. In this way, both contemporary advances and historically significant milestones were incorporated to provide a layered understanding of the field.

Priority was given to studies reporting empirical data on implant-embedded sensors, particularly electrochemical or biosensing platforms with relevance to orthopedic environments. Articles describing preclinical development, materials and signal optimization, and prototype validation were considered central to the scope. Clinical studies involving intraoperative or postoperative sensor systems—especially those applied to soft-tissue balancing, fixation analysis, or real-time monitoring—were included where they informed the context of biosensor integration. The review excludes purely wearable or external sensors, as well as purely theoretical or simulation-based models that did not demonstrate direct applicability to implant-based systems.

## 3. Technology Landscape

### 3.1. Implantable Sensor Modalities

Implantable biosensors have evolved dramatically over the last two decades, driven by the need for continuous, real-time physiological monitoring in hard-to-access anatomical sites. In the context of total knee arthroplasty (TKA), the ideal sensor must operate within the biomechanical constraints of the joint, remain functionally stable under dynamic load, and monitor either physical or biochemical changes without frequent recalibration. Several classes of sensors have been explored for orthopedic application, each offering unique advantages and limitations.

Mechanical and inertial sensors—such as accelerometers, gyroscopes, and strain gauges—have long been used to monitor kinematic parameters like gait, range of motion, and load distribution. These are commonly embedded in commercial smart implants such as the Persona IQ^®^ platform. However, while biomechanical data can reflect implant performance, it lacks the specificity required to detect early pathological changes, such as infection or osteolysis. This limitation has led to increasing interest in biochemical sensing modalities capable of monitoring the local environment of the prosthetic joint.

Electrochemical biosensors, in particular, offer compelling advantages. As reviewed by Yogev et al. [16], they enable the detection of specific ionic or molecular markers such as lactate, glucose, pH, or inflammatory cytokines by translating biochemical interactions into electrical signals. These sensors can be miniaturized, powered wirelessly, and integrated onto curved or flexible substrates suitable for orthopedic environments. Shanbhag et al. [17] emphasized the versatility of these systems, distinguishing between amperometric, potentiometric, and impedance-based sensors depending on the transduction mechanism.

Microneedle-based sensors have emerged as a minimally invasive sub-class within the implantable category. Designed originally for dermal interstitial fluid (ISF) access, these platforms are increasingly adaptable to orthopedic contexts. Kim et al. [18] and García-Guzmán et al. [19] highlight their ability to monitor ISF analytes such as glucose, pH, and electrolytes with high spatial specificity and minimal immune response. Lin et al. [20] further advanced this approach by integrating aptamer-based recognition into microneedle platforms for real-time pharmacokinetic monitoring, an innovation with direct relevance to infection tracking or drug-response monitoring in postoperative knees.

Organic electrochemical transistors (OECTs) also represent a promising class of soft, flexible sensor systems for integration into implants. Marquez et al. [21] demonstrated their capacity to amplify signals at low operating voltages, making them suitable for low-power applications. These devices operate through the coupling of ionic and electronic transport, offering both sensitivity and spatial adaptability—an essential requirement for conforming to the geometry of orthopedic implants.

In sum, the sensor modality landscape is rapidly diversifying, with platforms ranging from traditional inertial units to bioresponsive microneedles and transistor-based systems. The challenge lies not only in sensor selection but in determining which configurations offer the most clinically actionable data within the anatomical and functional constraints of the tibial stem or baseplate.

### 3.2. Electrochemical Sensing Principles

Electrochemical biosensors function by transducing biochemical interactions into electrical signals, enabling the detection of molecular events at extremely low concentrations and in real time. Their core appeal in orthopedic settings lies in their ability to continuously monitor clinically relevant biomarkers—such as pH, lactate, nitric oxide, or ions—directly at the implant–tissue interface, often before physical symptoms or imaging abnormalities arise. This makes them ideal candidates for integration into total knee arthroplasty (TKA) implants where early detection of complications is critical.

Fundamentally, electrochemical biosensors operate through three main transduction modes: amperometric, potentiometric, and impedimetric (or capacitive). Amperometric sensors measure current changes resulting from redox reactions at the electrode surface and are particularly suited for enzymatic detection of species like glucose, lactate, and hydrogen peroxide. These sensors benefit from high sensitivity but require careful design to maintain specificity and minimize drift over long implantation periods [22]. Potentiometric sensors, such as ion-selective electrodes (ISEs), measure voltage shifts due to changes in ionic concentrations. These are commonly applied in pH sensing and sodium/potassium monitoring, as detailed by Fan et al. [23], and are attractive for continuous infection surveillance. Impedance-based sensors track changes in interfacial charge properties and can be highly sensitive to biofilm formation or cellular deposition, although their application in orthopedics remains nascent.

Sensor performance is largely dictated by the recognition layer. Many biosensors rely on enzymes (e.g., lactate oxidase, glucose oxidase), aptamers, or ion-selective membranes to confer specificity. The use of direct electron transfer enzymes, as demonstrated by Gil et al. [24] for lactate detection in orthopedic infections, enhances signal clarity by reducing mediator interference. Similarly, Lin et al. [20] incorporated aptamer-functionalized microneedles to track circulating drug levels with high temporal resolution, pointing toward a future of personalized dosing feedback directly from within the joint.

Electrode architecture is another critical element. Designs range from planar electrodes to microband, interdigitated, or 3D nanostructured configurations. Pupel et al. [25] showed how gold nanobowl and nanourchin structures on graphene oxide substrates significantly enhanced hydrogen peroxide sensitivity due to increased surface area and optimized electron transfer. Such architectural tuning allows sensors to operate effectively at lower potentials, improving biocompatibility and reducing power demands.

Sensor selectivity and stability are enhanced through surface coatings and nanomaterial functionalization. For instance, Li et al. [26] developed a transient electrochemical sensor using Au nanomembranes coated with poly(eugenol) to selectively detect nitric oxide in joint cavities—demonstrating both biochemical fidelity and wireless data transmission. Judl et al. [27] clinically validated pH as a reliable infection biomarker, with a threshold of 7.4 offering over 95% specificity, reinforcing the need for high-fidelity pH biosensors at the prosthetic interface. Rich et al. [28] extended this concept by embedding pH, temperature, and strain sensors into biodegradable implants for continuous in vivo monitoring in large animals.

Importantly, electrochemical signal processing must account for biological noise, baseline drift, and environmental interference. Shanbhag et al. [17] and García-Guzmán et al. [19] stress the necessity of calibration algorithms, filtering, and compensation systems—especially when sensors are operating in variable joint environments. ISF-based microneedle sensors, for example, require correlation algorithms to map ISF analyte concentrations to blood-level equivalents, critical for actionable clinical interpretation.

Taken together, the electrochemical sensing principles that underpin orthopedic biosensors are rapidly maturing—from analyte-specific recognition chemistry to signal stabilization and real-time data communication. Their continued development is essential not only for integrating monitoring capabilities into the tibial component but for unlocking early, personalized orthopedic diagnostics. Figure 1 helps highlight the material-environmental interactions in implantable biosensors.

### 3.3. Materials and Fabrication Techniques

The success of biosensor-embedded tibial components in total knee arthroplasty (TKA) hinges not only on the sensing modality but on the materials used and the fabrication strategies employed to ensure long-term stability, precision, and biocompatibility under mechanical load. Unlike wearables or dermal patches, implantable biosensors in orthopedics must withstand cyclic stress, synovial fluid exposure, immune surveillance, and sterilization processes, all without delamination, degradation, or signal drift. The field has therefore invested significantly in developing robust sensor architectures and biointerfaces that can integrate seamlessly with load-bearing orthopedic hardware.

Electrode materials form the foundation of any electrochemical sensor. Common substrates include noble metals like platinum, gold, and silver/silver chloride, chosen for their conductivity and biostability. To enhance surface area and reduce overpotential, nanostructures—such as gold nanoparticles (Meng et al. [29]), graphene nanosheets, and carbon nanotubes—are frequently employed. These modifications improve both sensitivity and response time. Bocchetta et al. [30] further demonstrate the value of chitosan coatings and biopolymer layering for immobilizing enzymes and improving cell compatibility at the bio–electrode interface.

A key challenge is ensuring the mechanical and chemical compatibility of these sensor elements with the implant substrate—typically titanium alloys, UHMWPE, or PEEK. Researchers have pursued various strategies to address this. Markwitz et al. [31] explored the use of ion-implanted diamond-like carbon (DLC) coatings to improve surface conductivity and biocompatibility. Similarly, parylene coatings (Zeniieh et al. [32]; Kuppusami & Oskouei [33]) are widely recognized for their moisture barrier properties, pinhole-free deposition, and long-term biostability, making them ideal for encapsulating sensitive electronic components.

Substrate flexibility is especially relevant when sensors are printed or embedded onto non-planar surfaces like the tibial stem or tray. Fumeaux et al. [34] introduced degradable organic electrochemical transistors (OECTs) printed with PEDOT:PSS and carbon electrodes on flexible substrates, while Golabchi et al. [35] highlighted strategies for fluidic microchannel integration, signal conditioning circuitry, and low-power digital conversion in implantable systems. Additive manufacturing technologies—such as direct ink writing, inkjet, and screen printing—are also gaining traction for their ability to create custom sensor geometries with layer-by-layer precision [36].

Sensor encapsulation and packaging remain critical to device longevity. In orthopedic settings, this involves balancing protection from mechanical and biochemical degradation with permeability to target analytes. Materials like parylene-C, hydrogels, siloxane layers, and bioresorbable polymers (e.g., PLGA, PCL, PTMC) have been tested extensively [37,38]. Notably, biodegradable sensors are increasingly viewed as favorable for temporary monitoring windows, after which the system harmlessly degrades without surgical removal—an especially appealing approach for infection surveillance or early post-operative tracking.

The integration of multiple sensors into a single system—combining pH, strain, temperature, and motion detection—has also been successfully demonstrated. Rich et al. [28] achieved this in a large-animal model using flex-PCBs, ISFET-based pH sensing, and layered encapsulation for simultaneous monitoring, pointing to the feasibility of multi-analyte implants in orthopedic research. Meanwhile, Golabchi et al. [35] and Fan et al. [23] argue for modular sensor architectures capable of interfacing with cloud-based telemetry platforms or hospital monitoring systems—critical for clinical translation.

In summary, the materials and fabrication strategies driving orthopedic biosensor development are rapidly converging on solutions that prioritize miniaturization, resilience, sensitivity, and biocompatibility. The key challenge now lies in system-level integration: embedding these components into tibial platforms without compromising implant strength or surgical workflow, while ensuring reliable long-term function in vivo. Table 1 summarizes various technologies we discuss.

### 3.4. Electrochemical Characterization Protocols

A critical limitation of existing work on biosensor-integrated tibial components is the lack of detailed electrochemical testing protocols. While the principles of cyclic voltammetry (CV), chronoamperometry (CA), and electrochemical impedance spectroscopy (EIS) are often described, essential methodological details—such as potential windows, scan rates, frequency ranges, perturbation amplitudes, electrode configurations, and replicates—are rarely specified. This absence undermines reproducibility and makes it difficult to evaluate sensor performance rigorously [38,39,40].

Across the electrochemical literature, a consensus has emerged on standardized practices. Typical EIS protocols employ three-electrode setups, small perturbation amplitudes of 5–10 mV, and frequency sweeps spanning from sub-Hz to hundreds of kHz. Equivalent circuit modeling is used to extract parameters such as solution resistance, charge transfer resistance, and double-layer capacitance, while depressed semicircles or phase shifts are accommodated using constant phase elements and Gerischer terms (Fukase et al. [41]; Brett [42]; Lazanas & Prodromidis [43]). For CV and CA, scan rate selection is crucial for distinguishing between kinetic and diffusion control, and current response scales directly with analyte concentration. These methods provide the backbone for calibration curves and quantitative interpretation.

Concrete demonstrations underline the value of such protocols. Cholesterol biosensors, for example, achieved detection limits of 1.2 × 10^−7^ M using CV and 5.4 × 10^−7^ M using CA when properly parameterized (Grieshaber et al. [44]). In broader sensor platforms, the charge-transfer resistance measured via EIS has been shown to scale quantitatively with analyte concentration, though misapplication of equivalent circuits remains a common pitfall (Randviir & Banks [45]). Importantly, calibration must conform to international standards: IUPAC defines the limit of detection (LOD) as the mean of blank measures plus three times the standard deviation, and the limit of quantification (LOQ) as ten times the standard deviation (Mocak et al. [46]). These definitions justify the reviewer’s call for calibration curves with error bars and statistical rigor.

Recent advances in bio-integrated sensors illustrate the translational potential of these principles. In vivo monitoring platforms using CV, differential pulse voltammetry (DPV), and EIS now enable continuous detection of physiologically relevant analytes, including dopamine (1–106 nmol/L) and glutamate (40–60 μM), aided by antifouling coatings and wireless transmission modules [38]. Similarly, next-generation bioelectrodes incorporating enzymatic, aptamer, and CRISPR-based interfaces demonstrate the breadth of electrochemical sensing modalities applicable to load-bearing implants [47,48]. Collectively, these studies establish clear methodological expectations: orthopedic biosensor research must report full electrochemical protocols, provide calibration data with statistical context, and align with international standards to ensure reproducibility and translational relevance. Table 2 summarizes different recent findings in accordance with new biosensor research findings. Figure 2 illustrates the calibration of these electrochemical sensors.

## 4. Integration into Tibial Components

### 4.1. Mechanical and Structural Considerations

Embedding biosensors within the tibial component of a total knee arthroplasty (TKA) requires a delicate balance between mechanical integrity, sensing accuracy, and spatial economy. The design must preserve the implant’s structural strength while ensuring robust sensor function, often within the constraints of the tibial tray or stem.

Early experimental models by Roth et al. [49] introduced a sensor-integrated tibial baseplate that allowed full in vitro assessment of tibiofemoral forces. The design preserved the tray’s biomechanical fidelity while supporting accurate force localization. Similarly, cadaveric validation studies by Al-Nasser et al. [50] showed that embedding force sensors in the Persona system’s tray did not compromise joint line integrity or positioning across a variety of insert thicknesses. These findings demonstrate that embedded sensors can maintain anatomical fidelity when appropriately integrated.

Hall et al. [51] further advanced this approach by developing a battery-free piezo-coil debonding detection system placed within a tibial groove. Their sensor architecture did not alter the external morphology of the implant, affirming that discreet sensor integration is feasible without disrupting the prosthesis profile or articulating surfaces. These insights were echoed in broader system reviews, such as Wu et al. [52], who emphasized that system-level integration—including circuit routing and signal conditioning—must be optimized for the curved, load-bearing surfaces characteristic of knee implants.

In addition to architectural compatibility, materials used in encapsulating sensor systems must withstand the joint’s mechanical stresses. Kohler et al. [53] subjected parylene-C encapsulated piezoelectric actuators to over 500,000 loading cycles, confirming durability under prolonged biomechanical stress. Similarly, Feig et al. [54] demonstrated that biodegradable electronic platforms fabricated on PLGA and PCL could endure mechanical flexing while undergoing controlled resorption, making them suitable for temporary, post-operative sensing roles.

These considerations are further complicated when sensors are fabricated using flexible or printed electronics. Boček et al. [55] successfully fabricated Prussian blue-modified graphene sensors on polyimide substrates using photonic sintering, producing highly conductive, chemically stable electrodes. Their work suggests that even on curved surfaces like the tibial tray, printed sensors can achieve both conformance and functional stability.

Ultimately, structural design must support signal fidelity without compromising the implant’s mechanical strength or its fixation to the host bone. The use of modular compartments, mechanical grooves, and custom stem cavities are emerging as practical solutions. Rich et al. [28] exemplified this with their successful integration of pH, strain, and temperature sensors into a live animal model, where implant geometry and sensing capability were harmonized through careful spatial planning.

Beyond architectural integration, durability under cyclic loading is critical for translational feasibility. ISO 14879-1 defines tibial tray endurance using a cantilever-based cyclic loading protocol, with ten million cycle runout serving as a benchmark for survival [56]. ASTM F1800 likewise establishes constant-amplitude fatigue testing for metallic trays, providing reproducible comparisons though not fully predictive of clinical outcomes [57]. EndoLab standards (ISO 14243, ASTM F1223, F2722, F2777) expand this framework to include constraint and high-flexion scenarios, ensuring cross-laboratory consistency [58].

Simulation studies highlight how in vivo conditions can exceed these baselines. Mell et al. [59] demonstrated that femoral center of rotation shifts can increase wear by up to 35%, while Dreyer et al. [60] showed in vivo wear rates nearly triple those predicted by ISO simulators, with altered wear scar locations. These findings show that while standardized endurance tests remain indispensable for regulatory compliance, physiologically relevant kinematics are required to evaluate sensorized tibial implants realistically. Figure 3 highlights the structure of a typical biosensor we refer to. 

### 4.2. Powering and Data Transmission

Powering embedded biosensors presents one of the greatest engineering challenges for smart TKA systems. Given the limited volume within the tibial component, traditional batteries are often unsuitable due to bulk, thermal risk, and long-term degradation. As a result, wireless energy strategies—such as inductive, capacitive, and acoustic energy transfer—are gaining traction.

Khan et al. [61] provided a comprehensive evaluation of these approaches, highlighting inductive power transfer as the most viable for orthopedic applications. Their review emphasized that coil geometry, alignment, and thermal output must be optimized to avoid soft-tissue heating and metal-induced attenuation. Similarly, Kim et al. [62] detailed the constraints associated with leadless, batteryless biosensor platforms, noting that implant geometry, specific absorption rate (SAR), and electromagnetic shielding must all be considered during design.

Wireless power systems must also accommodate reliable data transmission—a requirement that introduces trade-offs in range, latency, and energy cost. Technologies such as Bluetooth Low Energy (BLE), near-field communication (NFC), and radiofrequency (RF) telemetry each offer different advantages. Wu et al. [52] outlined a flexible communication strategy based on device location and power availability. NFC systems, for instance, are effective for short-range, high-efficiency transmission and have already been applied in biosensor prototypes implanted within joint cavities [62,63]. BLE systems, while more power-hungry, allow for greater mobility and may be suitable in scenarios where the implant is paired with external wearable relays.

Signal shielding, reflection within metallic enclosures, and the proximity of tissue–bone interfaces all introduce interference risks. Rich et al. [28] addressed this challenge by employing flex-PCBs shielded by biocompatible encapsulants, enabling continuous signal transmission over extended periods in vivo.

The convergence of wireless powering and telemetry has opened the door to truly autonomous implant systems. However, these strategies must be tailored to the unique electrical, anatomical, and mechanical properties of the tibial component—underscoring the need for orthopedics-specific design frameworks.

### 4.3. Sterilization and Surgical Workflow

The final consideration in biosensor integration is surgical compatibility, particularly how sensors and their encapsulating materials perform under clinical sterilization protocols such as autoclaving or gamma irradiation. In a recent study, Mohamed [64] investigated the impact of repeated autoclaving on orthopedic instruments and found significant degradation in fatigue resistance and surface microstructure after multiple cycles. These findings highlight the need for sensor packaging materials that can endure both thermal and radiological sterilization without compromising structural or sensing function.

Encapsulation materials such as parylene-C, used successfully by Kohler et al. [53], offer resistance to moisture, oxidation, and temperature extremes, making them well-suited for sterilization-resistant coatings. Similarly, Kohler et al. [53] reviewed parylene variants and their mechanical and chemical stability under sterilization conditions, noting Parylene-N and Parylene-C as preferred for implantable systems.

Beyond sterilization, sensorized implants must not interfere with surgical workflows, including instrumentation, alignment tools, and operative timelines. In this regard, designs like those by Hall et al. [51] are exemplary, integrating wireless sensing into tibial grooves without altering surface geometry or requiring additional surgical steps. Al-Nasser et al. [50] similarly demonstrated seamless integration into conventional tray systems, ensuring that the presence of sensors did not affect intraoperative balancing or insert placement.

Future integration frameworks may require alignment with smart OR technologies, enabling intraoperative diagnostics, sensor function validation, or automated alerts. These possibilities, though exciting, must be balanced with the current demands of sterility assurance, tool compatibility, and operative efficiency. Table 3 summarizes several procedural strategies from the literature.

## 5. Preclinical and Clinical Evidence

Biosensor-enabled tibial components are supported by a growing body of preclinical work and early clinical experience. Collectively, these studies confirm feasibility, safety, and translational potential, while underscoring the need for larger, long-term trials.

### 5.1. In Vitro and Animal Models

A strong foundation of preclinical and in vitro studies has validated the safety, durability, and performance of biosensor-integrated tibial components. Early rodent models with nitric oxide sensors demonstrated accurate inflammatory monitoring and full biodegradation without immune reaction [62]. Rich et al. [28] advanced this by embedding a suite of pH, temperature, and strain sensors into tibial implants in a large-animal model, which maintained structural integrity and delivered continuous data for several weeks, confirming feasibility of multi-analyte sensing in load-bearing joints.

Energy autonomy has also been investigated. Ibrahim et al. [65] built a triboelectric nanogenerator harvesting gait energy, while Liu et al. [66] reviewed biodegradable piezoelectric materials (PLLA, PVDF, ZnO composites) that can power themselves while degrading safely. In chemical sensing, Ding et al. [63] validated a wireless lactate biosensor in an ex vivo bone model for rapid infection detection.

Comprehensive reviews (Ding et al. [63]; Wu et al. [52]) emphasize miniaturization, flexible architectures, and wireless telemetry as essential to stable performance. Together, these preclinical and benchtop studies establish that biosensors can be embedded, powered, and functionally integrated without compromising implant mechanics (Table 4).

### 5.2. Human Pilot Studies, RCTs, and Clinical Translation

Clinical investigations have focused on two domains: intraoperative balancing systems and postoperative smart implants.

Intraoperative load sensors: such as VERASENSE and eLIBRA quantify compartmental loads in real time, allowing surgeons to adjust ligament balance toward predefined thresholds. RCT evidence confirms these devices improve intraoperative balance: Wood et al. [67] found only 5.3% of sensor-guided knees remained unbalanced vs. 35.5% in controls, while Afshari et al. [68] replicated this in a multicentre RCT (unbalanced 9.4% vs. 36.8%), with modest improvement in walking distance but no long-term PROM advantage. A systematic review by Sava et al. [69] corroborated these findings, noting lower manipulation-under-anesthesia (MUA) rates but no consistent differences in ROM, PROMs, or reoperation. Cho et al. [70] similarly reported improved ligament balancing intraoperatively but inconclusive long-term impact. The overall picture is clear: load sensors deliver reproducible balance improvements, but sustained clinical superiority remains unproven.

Smart tibial implants: Persona IQ represents the first FDA-cleared “smart” tibial component, embedding an inertial measurement unit for continuous gait telemetry. Case-level evidence suggests feasibility and early functional benefit: Cushner et al. [71] reported detection of delayed recovery trajectories, enabling timely intervention, while Yocum et al. [72] described a series of three patients achieving KOOS-JR > 90 at three months, with >95% data transmission fidelity and ~2× gait-speed improvements. Design reviews (Guild et al. [73]) describe robust onboard telemetry with battery life projected to ~10 years, though constrained by stem length and periodic capture protocols. While early experience demonstrates safety and practicality, larger controlled trials are essential to link telemetry signatures with hard outcomes such as revision or reoperation (Table 5).

## 6. Challenges and Knowledge Gaps

Despite promising advancements, the integration of biosensors into tibial components for total knee arthroplasty (TKA) faces persistent challenges across technical, biological, regulatory, and ethical domains. These barriers must be addressed to enable safe, effective, and scalable clinical adoption.

### 6.1. Technical Hurdles

Biosensor function is fundamentally constrained by the reliability of the embedded hardware and its behavior in physiological environments. A central challenge is signal drift and instability, often driven by the degradation of sensing materials and exposure to complex biofluids. Youssef et al. [74] highlighted these vulnerabilities in a review of pH, temperature, and oxygen sensors used in chronic wound monitoring, noting that signal degradation, calibration drift, and delayed response times undermine real-time performance—concerns that directly extend to tibial implant systems. The electronic architecture of many biosensors relies on organic semiconductors such as PEDOT:PSS, which offer low-voltage operation and high sensitivity but suffer from conductivity loss and instability in wet, mechanically dynamic environments (Wang et al. [75]). Mechanical–electrical interfaces are likewise prone to fatigue, microfractures, and delamination under repetitive load cycles. Without robust encapsulation strategies that balance flexibility, insulation, and biostability, long-term sensor function remains unreliable. Furthermore, the absence of in situ calibration frameworks limits precision as the implant environment drifts physiologically over time.

Corrosion and electrochemical stability represent additional hurdles with direct implications for sensor lifetime. Standardized cyclic potentiodynamic polarization testing (ASTM F2129) is widely used to evaluate the susceptibility of implant alloys and coatings to localized corrosion and pitting in simulated physiological electrolytes. While intentionally aggressive, these tests remain the regulatory benchmark [76]. Electrochemical noise (EN) analysis, performed under zero-resistance ammetry conditions, offers complementary insight by capturing voltage and current fluctuations associated with pit initiation, though interpretation requires careful detrending to reconcile differences from polarization resistance obtained by EIS (BioLogic [77]). Together, these methods establish the minimum expectations for corrosion characterization of implantable biosensors.

Immersion–fatigue studies provide further evidence of how chemical degradation and mechanical cycling interact in implant environments. Additively manufactured irons exhibit longer incubation before corrosion onset but reduced fatigue strength compared to hot-rolled controls, with microstructural defects driving early pit formation (Wackenrohr et al. [78]). Nanoparticle reinforcement alters this balance: Fe_2_O_3_ additions reduce fatigue resistance, whereas CeO_2_ improves fatigue properties but accelerates degradation (Wackenrohr et al. [79]). Comparable results are seen in magnesium alloys, where PLA coatings initially extend fatigue life but undergo delamination and pit-driven failure after immersion in concentrated SBF, reducing strength by up to 35% (Talesh & Azadi [80]). These findings emphasize the need to combine electrochemical assays with high-cycle fatigue protocols that mimic joint mechanics.

Polymer-based electrodes and hydrogels also present stability limitations. Photopatternable PEDOT:PSS hydrogels have been shown to retain conductivity (~30 S/cm) and mechanical flexibility (~50% strain) under aqueous immersion (Wang et al. [81]), while hybrid crosslinked hydrogels preserved nearly 90% conductivity after 1000 cycles of loading (Li et al. [82]). Although these materials show promise, their long-term performance remains dependent on systematic evaluation of fouling, hydrolysis, and electrochemical drift in simulated body fluids.

Taken together, these findings demonstrate that the technical hurdles in biosensor-integrated tibial components extend beyond signal drift and encapsulation design to include well-documented risks of corrosion, electrochemical instability, and mechanical-chemical coupling. A rigorous framework combining cyclic polarization, EN and EIS monitoring, immersion–fatigue testing, and polymer stability screening is required to ensure scientific robustness and translational reliability (Table 6). Figure 4 illustrates the cyclic potentiodynamic polarization and electrochemical noise context of this technology.

### 6.2. Biological Constraints and Biocompatibility

Biosensor-integrated tibial components must operate within a biologically hostile environment that resists foreign materials. The foreign body reaction (FBR) remains the most formidable barrier to implant longevity. Anderson et al. [83] demonstrated how macrophages and fusion-derived giant cells initiate chronic inflammation and fibrous capsule formation, which progressively isolates sensors from their analytes. This encapsulation process effectively silences devices over time. Li et al. [84] further showed that implant stiffness modulates fibroblast and macrophage activity, exacerbating fibrosis when substrates lack biomechanical compatibility. These responses cannot be averted by targeting a single pathway: Sharon et al. [85] found that even with microglial depletion, astrocyte-driven fibrosis persisted, underscoring the redundancy of inflammatory cascades against implants.

Material innovations have turned toward antifouling and immune-modulatory coatings. Wang et al. [86] reported that zwitterionic polymers outperform PEG and PHEMA by reducing protein adsorption and microbial colonization, thereby delaying capsule formation. Such coatings illustrate how surface chemistry can extend biosensor lifespan.

In parallel, standardized cytotoxicity and hemocompatibility testing frameworks provide quantitative assurance of material safety. ISO 10993-5 defines cytotoxicity thresholds (≥70% cell viability), while ISO 10993-4 and ASTM F756 address hemocompatibility through hemolysis, coagulation, platelet activation, and complement assays. FDA’s ASCA pilot explicitly recognizes these standards as part of the required safety panel for implantable devices [87,88,89].

Recent studies confirm that the key materials used in tibial biosensors meet these standards. PEDOT:PSS and polypyrrole achieve >90% viability in MEM elution assays when processed to remove residual solvents. Graphene and CNT composites report hemolysis <2% with negligible complement activation, qualifying as non-hemolytic. Parylene-C, widely used for encapsulation, consistently demonstrates >95% cell viability and <2% hemolysis, while also suppressing leachables from polymer substrates and sustaining over 500,000 loading cycles without degradation (Kohler et al. [53]; Menzel et al. [90]). PEDOT:PSS hydrogels extend this profile by combining high conductivity with mechanical compliance to tissue modulus ranges, thereby minimizing inflammatory activation (Li et al. [82]).

The most relevant materials for tibial biosensor integration comply with ISO/ASTM requirements and exhibit strong biological performance. Nonetheless, compliance does not eliminate the FBR, which remains an inevitable challenge. Long-term functionality will therefore require combined strategies—mechanical compliance, standardized biocompatibility, and antifouling surface chemistries—to mitigate encapsulation and sustain analyte access (Table 7).

### 6.3. Tribology and Tribocorrosion

Beyond structural fatigue, biosensor-integrated tibial implants must also withstand tribological stresses at articulating and sensor–tissue interfaces. Wear, friction, and tribocorrosion act synergistically, accelerating degradation and jeopardizing long-term sensor function.

Protein interactions at the implant surface can either stabilize oxide layers or promote breakdown depending on environmental conditions. Takadoum [91] highlighted that titanium alloys alloyed with Nb, Zr, or Ta show enhanced resistance compared to Ti–6Al–4V, yet tribocorrosion processes remain a contributor to implant loosening and impaired osseointegration. Puthillam & Selvam [92] reviewed the combined effects of adhesive, abrasive, corrosive, fretting, cavitation, and oxidative wear, showing that protein–peroxide interactions accelerate corrosion of Ti alloys and alter open circuit potentials. Such findings reinforce that wear cannot be considered in isolation from electrochemical degradation.

Experimental evidence supports these mechanistic insights. Baykal et al. [93] demonstrated that multidirectional pin-on-disk testing more accurately reproduces clinical wear than unidirectional protocols, with cross-shear dramatically increasing UHMWPE wear. Highly crosslinked UHMWPE reduced wear by nearly an order of magnitude compared to conventional grades, showing the importance of material selection. Dreyer et al. [60] confirmed that in vivo wear rates significantly exceed ISO simulator predictions, emphasizing the gap between bench protocols and real joint mechanics.

Tribocorrosion is a decisive hurdle for biosensor integration, where coatings, electrodes, and encapsulation layers must withstand not only cyclic load but also protein-mediated electrochemical attack (Figure 5). Standardized wear and fatigue tests remain necessary for regulatory approval, but advancing tribocorrosion protocols that incorporate proteins, peroxides, and multidirectional motion will be essential to demonstrate realistic durability. Table 8 highlights key several key testing approaches.

### 6.4. Operational Stability and Antifouling Strategies

The operational lifespan of biosensor-integrated tibial implants is threatened less by mechanical failure than by biological fouling and electrochemical drift. Synovial fluid contains high concentrations of proteins, enzymes, and cellular components that rapidly adsorb onto electrode surfaces, blocking analyte access and shifting baseline signals. In parallel, fluctuations in pH, ionic strength, and temperature introduce calibration instability. These challenges compromise long-term signal fidelity and remain major barriers to clinical translation.

Protein fouling: PEG-based films, once considered the standard for antifouling, provide only short-term protection; they degrade within 72 h in serum-rich media (Jarosińska et al. [94]). In contrast, zwitterionic polymers such as poly(sulfobetaine methacrylate, pSBMA) and PSBEDOT resist nonspecific adsorption by forming stable ionic hydration layers. Yang et al. [95] demonstrated that PDA-PSB coatings reduced protein adsorption by 89% and fibroblast adhesion by 86%, markedly decreasing inflammatory activation in vivo. Similar findings have been reported for zwitterionic PEDOT derivatives, which maintain linear electrochemical responses even in 100% plasma.

Hybrid and biomimetic strategies: PEDOT/PEG hydrogel composites maintain >90% of current response in undiluted serum (Campuzano et al. [96]), combining antifouling with conductivity. Zwitterionic peptides immobilized on PEDOT interfaces support femtomolar DNA detection in plasma while resisting fouling (Cui et al. [97]). Ye et al. [98] extended this approach to nucleic acid nanostructures, showing stable microneedle sensor operation for up to 14 days in vivo. These examples underscore the growing emphasis on dual-function coatings that provide both analyte recognition and fouling resistance.

Electrode anchoring: Beyond surface coatings, bonding chemistry significantly affects stability. Liu et al. [99] reported that Au–C≡C and bidentate thiols reduced electrode drift to <6% in glutathione-rich environments, compared with ~23% for conventional Au–S bonds. When paired with antifouling coatings, such chemistries ensure more durable interfaces in redox-active fluids like synovial joint environments.

Environmental effects: Physiological variations in pH and ionic strength generally cause only modest error (<20%), but temperature fluctuations produce major drift. Fetter et al. [100] found that increases of just 7–8 °C induced signal errors up to 63%, demonstrating that real-time temperature correction is indispensable for reliable in vivo sensing.

Thus PEG alone seems inadequate for chronic tibial applications. Instead, future biosensors must integrate zwitterionic polymers, biomimetic peptides, conductive hydrogels, and robust electrode anchoring chemistries, alongside environmental compensation mechanisms. These strategies are complementary: where coatings prevent nonspecific fouling, electrode design enhances stability, and real-time correction addresses unavoidable physiological fluctuations (Table 9).

### 6.5. Regulatory & Standards

Current regulatory frameworks lag behind the pace of biosensor innovation. Neither the FDA nor EMA has yet defined a clear, streamlined pathway for active orthopedic implants containing integrated biosensing platforms. The, U.S. FDA guidance on electromagnetic compatibility (EMC) for medical devices emphasizes emissions testing, immunity thresholds, and risk management, but lacks specificity for implants that collect and transmit continuous biochemical data [101,102].

Similarly, ISO 14708-1 outlines baseline requirements for safety, packaging, and performance in active implantable devices, but does not address sensor calibration, telemetry security, or multi-analyte integration [103]. These omissions represent major roadblocks to biosensor-equipped TKA systems reaching regulatory clearance.

Camara et al. [104] underscored the risks posed by insufficiently protected implantable medical device (IMD) communication systems, including telemetry hijacking, battery drain, and unauthorized data extraction. Without embedded encryption protocols and authenticated access control, biosensor implants remain vulnerable to interference and data misuse.

Moreover, device manufacturers must navigate fragmented approval requirements, long timelines, and evolving standards—challenges that disincentivize innovation and delay translation into practice.

### 6.6. Ethical & Data Considerations

Finally, the use of biosensor-enabled implants introduces a new layer of ethical complexity, particularly regarding data governance, patient autonomy, and equity.

At the center lies the issue of data privacy and ownership. Vayena et al. [105] emphasized that digital health ecosystems frequently operate without robust consent models, often using real-world patient data for purposes beyond clinical care. This raises concerns about transparency, data sharing, and secondary use—especially for continuously streaming sensor data embedded in patients’ bodies.

Kaur et al. [106] reviewed vulnerabilities in wireless wearable sensor networks (WWSNs), identifying persistent risks of data leakage, cloud exposure, and lack of standard cybersecurity protocols. These vulnerabilities apply directly to biosensor implants that rely on wireless telemetry, underscoring the need for pre-integrated security architecture.

Beyond privacy, justice and inclusivity are emerging as frontline concerns. Brall et al. [107] warned that digital health tools often fail to address digital illiteracy and socioeconomic barriers, potentially exacerbating health disparities if smart implants are deployed without accessible support systems.

Anticipating future applications, Hansson [108] explored the ethical implications of sensory prostheses and bio-enhancement, raising concerns about informed consent, identity shifts, and potential misuse of enhancement technology. These insights are particularly relevant as biosensor implants become more autonomous and potentially predictive, blurring the lines between therapy, surveillance, and augmentation.

Siala and Wang [109] proposed the SHIFT framework (Sustainability, Human-centeredness, Inclusiveness, Fairness, Transparency) as a model for embedding responsible governance into AI-driven digital health. Applying this framework to biosensor platforms could help align future implant systems with ethical expectations from both clinicians and patients.

In sum, while biosensor-integrated tibial components represent a compelling frontier in precision orthopedics, their implementation is constrained by fundamental technical instability, inflammatory interference, underdeveloped regulatory guidance, and significant ethical tensions. Addressing these gaps is not optional; it is essential to realizing the full potential of sensor-embedded TKA and ensuring its acceptance, safety, and impact at scale.

## 7. Future Roadmap & Recommendations

The integration of biosensor-enabled tibial components into total knee arthroplasty (TKA) represents a transformative frontier in orthopedic precision medicine. Yet to fully realize this potential, a multidimensional roadmap is required, spanning collaborative design, standardization, translational acceleration, commercial strategy, and next-generation innovations.

### 7.1. Interdisciplinary Collaboration

The complexity of biosensor-integrated implants demands a shift from isolated innovation to deeply interdisciplinary collaboration. Engineers, clinicians, data scientists, and regulatory strategists must operate within joint innovation ecosystems. Müller et al. demonstrated how finite element modeling of piezoelectric actuators, embedded in orthopedic hardware, can drive therapeutic personalization—provided that clinical workflows and simulation parameters are co-designed [110]. Similarly, Su et al. (2023) and Caprari et al. (2018) emphasized the role of educational and team-based models in bridging engineering with orthopedic care, recommending formal integration of biomedical design within surgical teams [111,112].

Historically, this synergy was present in the early conceptualization of smart implants. Burny et al. (2000) outlined a design philosophy grounded in shared language and iterative feedback between clinicians and technical developers—an approach that remains foundational [113].

### 7.2. Standardized Testing Protocols

Progress in biosensor design is currently limited by the lack of harmonized testing protocols. Variability in in vitro models, bench simulations, and preclinical trial design introduces irreproducibility and delays translation. Frisch et al. (2023) reviewed emerging 3D biomaterial testing systems that offer higher fidelity over traditional animal models, advocating for their wider adoption [114].

From a data and device interoperability perspective, Giorgi and Tonello (2022) called for alignment with frameworks such as ISO/IEC/IEEE 21451, which standardize transducer interfaces across wearable and implantable devices [115]. Biosensor implants for TKA must eventually comply with such frameworks to enable modular upgrades, reduce validation costs, and streamline cross-platform compatibility.

### 7.3. Translational Pathway

Despite promising prototypes, most biosensor implants remain trapped in the so-called valley of death between laboratory success and clinical adoption. Seyhan (2019) identified this gap as a result of fragmented funding, lack of translational strategy, and insufficient infrastructure to scale from feasibility to randomized trials [116].

To cross this divide, a three-phase translational pipeline is necessary: (1) preclinical and pilot testing with strong regulatory input, (2) multicenter feasibility trials with economic modeling, and (3) industry-academic consortia for market deployment. Andreu-Perez et al. (2015) stressed the importance of embedding biosensors into a broader clinical ecosystem, linking patients, devices, data streams, and decision-making platforms in a cohesive loop [8]. In addition, Scholten and Meng (2018) emphasized stable sensor performance as the gating variable for successful implantable closed-loop systems, underscoring the importance of translational reliability at every stage [117].

### 7.4. Commercialization & Reimbursement

The path to sustainable biosensor deployment will depend on payer alignment and commercial feasibility. Kelmers et al. (2023) outlined the economic and technological challenges unique to smart orthopedic implants, including high development costs, uncertainty in insurance coverage, and the lack of consensus on value-based evaluation criteria [15].

For reimbursement success, biosensor implants must demonstrate cost-effectiveness, clinical utility, and improved outcomes. This involves rigorous health technology assessment (HTA) and early interaction with payers during trial design. Mazzocchi (2016) detailed the strategic steps toward commercialization, including prototyping under regulatory constraints, navigating ISO/FDA pathways, and positioning for postmarket surveillance [118].

Cicha et al. (2022) further identified opportunities in chronic disease management where biosensor-driven implants could reduce readmission and enable outpatient management—broadening their use beyond the operating room and into longitudinal care [119].

This translational journey is summarized in Figure 6, highlighting the phased development, associated challenges, and necessary clinical transitions for biosensor-integrated tibial components.

### 7.5. Emerging Frontiers

Looking ahead, the frontier of biosensor-enabled implants lies in multi-analyte detection, AI integration, and closed-loop therapeutic feedback. Chen et al. (2025) [22] showcased how artificial intelligence can amplify biosensor value by enabling adaptive diagnostics, anomaly detection, and real-time personalization. These technologies not only monitor but also learn—transforming static implants into intelligent therapeutic agents [22].

Flynn and Chang (2024) described how point-of-care biosensing and AI convergence offers novel approaches for chronic monitoring, though they caution against algorithmic opacity and ethical drift [120].

The future also envisions biohybrid and biodegradable systems, as presented by Abyzova et al. (2023) [121]. These smart implants combine sensing, drug delivery, and energy harvesting through resorbable materials and wireless communication. These innovations reflect a shift from implant-as-hardware to implant-as-interface—enabling therapeutic and diagnostic feedback in one dynamic device.

## 8. Conclusions

The integration of biosensors into tibial components for total knee arthroplasty represents a compelling advancement in the evolution of orthopedic implants. From intraoperative load balancing to long-term infection monitoring and gait analysis, biosensor systems hold the potential to make prosthetic joints responsive, intelligent, and deeply personalized.

This review has shown that while preclinical and early clinical evidence supports feasibility, the field is still constrained by multiple systemic challenges, including technical degradation, immune interference, inadequate regulatory guidance, and unresolved data ethics. Addressing these issues requires an interdisciplinary, standards-driven, and ethically grounded roadmap.

The future of TKA is not merely mechanical, it is informational. As biosensors evolve to support multi-analyte detection, real-time AI integration, and closed-loop therapeutics, the tibial component may become not just a support structure, but an active partner in recovery, diagnostics, and rehabilitation. The success of this evolution depends not only on what we can build, but how, and with whom, we build it.

## Figures and Tables

**Figure 1 bioengineering-12-00988-f001:**
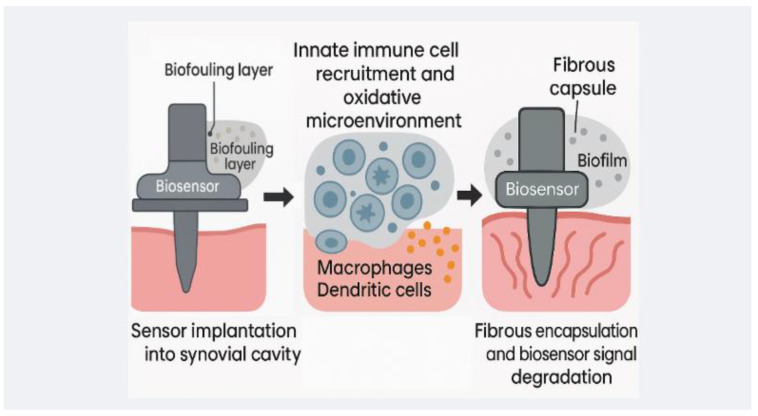
Schematic illustration of material-environment interactions in implantable biosensors, highlighting critical challenges such as immune response, encapsulation, and signal interference. Such dynamics are central to biosensor design in total knee arthroplasty applications.

**Figure 2 bioengineering-12-00988-f002:**
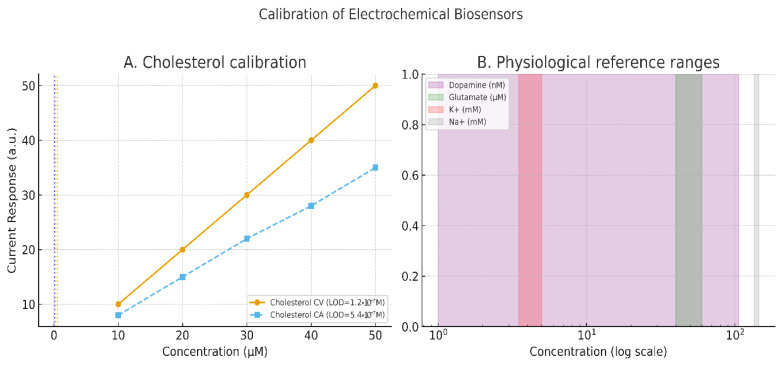
Panel (**A**) shows representative calibration curves for cholesterol detection using cyclic voltammetry (CV) and chronoamperometry (CA), illustrating differences in sensitivity and detection limits (CV: 1.2 × 10^−7^ M; CA: 5.4 × 10^−7^ M). Panel (**B**) highlights physiological concentration ranges for key biomarkers relevant to implantable biosensors, including dopamine (1–106 nmol/L), glutamate (40–60 μmol/L), potassium (3.5–5 mmol/L), and sodium (135–145 mmol/L). Together, these examples demonstrate the importance of reporting calibration data and contextualizing biosensor performance within physiologically relevant ranges.

**Figure 3 bioengineering-12-00988-f003:**
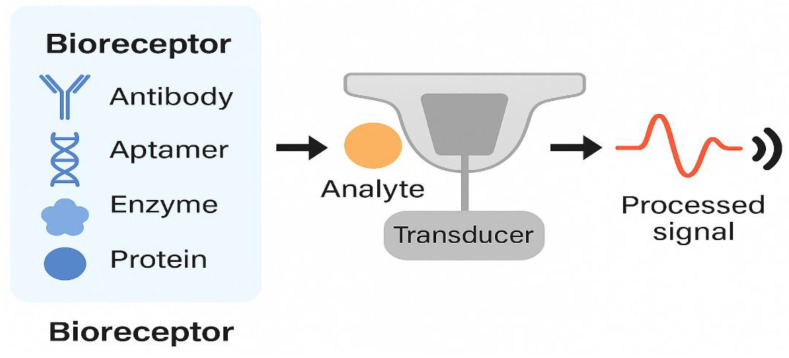
Schematic structure of a biosensor showing core components—bioreceptor layer, transducer element, and signal processor—functioning as a unified detection and transmission unit. This modular architecture forms the basis for integration into smart tibial implants.

**Figure 4 bioengineering-12-00988-f004:**
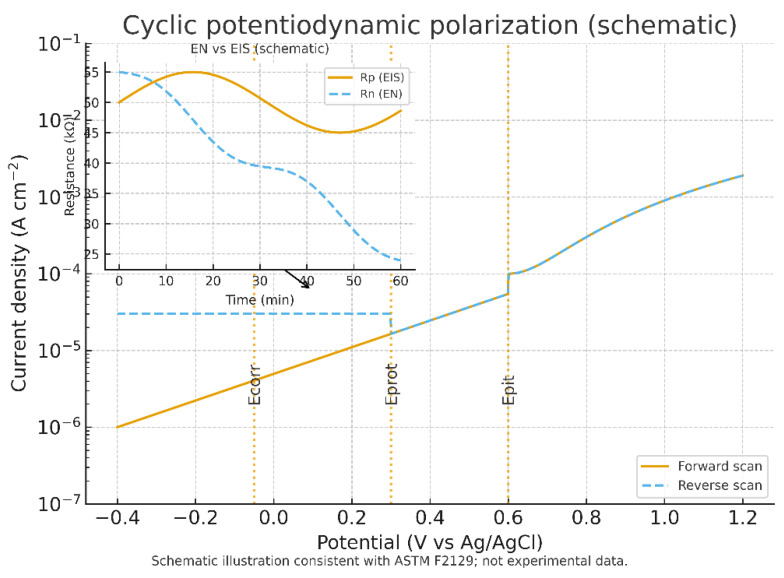
Cyclic potentiodynamic polarization and electrochemical noise context. The forward and reverse scans illustrate a typical ASTM F2129 curve, with corrosion potential (Ecorr), passive region, pitting potential (Epit), and protection potential (Eprot) annotated. The inset contrasts polarization resistance from EIS (Rp) with noise resistance from electrochemical noise measurements (Rn), highlighting that Rn can decline earlier during pit initiation. Schematic illustration consistent with ASTM F2129; not experimental data.

**Figure 5 bioengineering-12-00988-f005:**
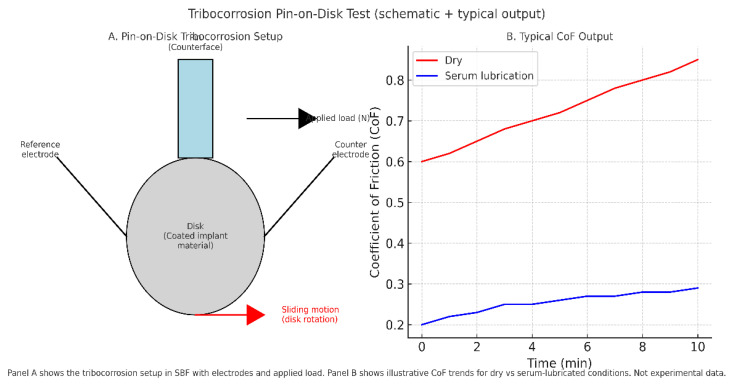
Tribocorrosion testing. Panel (**A**) shows a pin-on-disk setup, where a counterface pin applies load against an implant material disk submerged in simulated body fluid (SBF), with reference and counter electrodes monitoring electrochemical response. Panel (**B**) illustrates typical coefficient of friction (CoF) trends, with dry conditions producing progressive CoF increase and serum lubrication stabilizing CoF at lower values. Schematic illustration; not experimental data.

**Figure 6 bioengineering-12-00988-f006:**
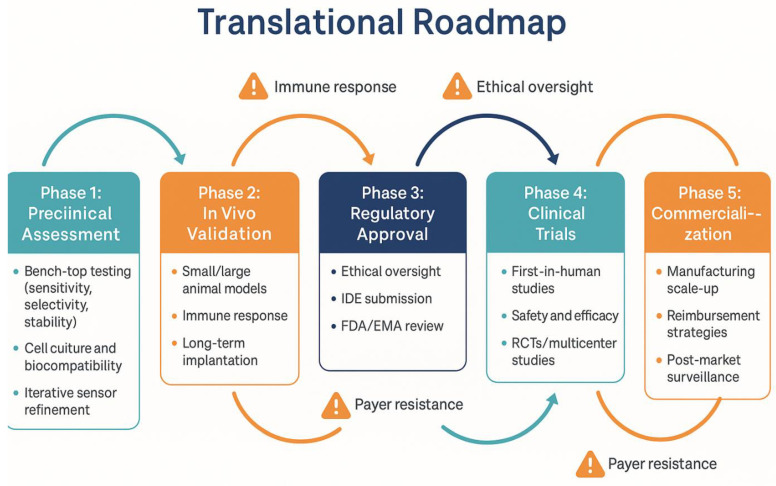
Translational roadmap for biosensor-enabled tibial components in TKA. The figure outlines five sequential phases: preclinical assessment, in vivo validation, regulatory approval, clinical trials, and commercialization, along with representative activities and major barriers (such as immune response, ethical oversight, payer resistance). The roadmap emphasizes a recursive and interdisciplinary approach essential for successful translation from bench to bedside.

**Table 1 bioengineering-12-00988-t001:** Summarizes the core sensing technologies, target biomarkers, and integration strategies relevant to biosensor-embedded tibial components in TKA.

Technology/Material	Function/Biomarker	Integration Strategy
Amperometric, Potentiometric, Impedance Sensors	Lactate, pH, NO, ions (Na^+^, K^+^, Clâˆ’)	Microelectrodes, nanostructured surfaces, enzyme/aptamer functionalization, ISFETs
Microneedle-based Sensors	Interstitial fluid analytes (glucose, lactate, electrolytes)	Minimally invasive dermal/tissue insertion, wearable or semi-implantable formats
Organic Electrochemical Transistors (OECTs)	Low-voltage signal amplification, glucose, dopamine, DNA	Printed PEDOT:PSS on degradable/flexible substrates; gate-modulated sensing
Gold Nanoparticles, Graphene, Carbon Nanotubes	Signal amplification, improved analyte detection sensitivity	Electrode surface modification, nanostructuring for conductivity and surface area
Chitosan and Biopolymer Coatings	Enzyme immobilization, cell adhesion, anti-fouling layers	Layer-by-layer deposition, integration with carbon-based electrodes
Parylene-C and Biodegradable Encapsulation Layers	Barrier protection, moisture sealing, signal insulation	Conformal coating of implant electronics; encapsulation of PCB and microfluidics
ISFETs and Flex-PCBs	Multi-analyte detection (pH, strain, temperature)	Embedded in titanium housing, soft packaging for strain/motion resistance
Additive Manufacturing (Inkjet, Direct Ink Writing)	Customizable electrochemical sensor geometries	Printed electrode systems on curved or layered implant surfaces

**Table 2 bioengineering-12-00988-t002:** Electrochemical Characterization Parameters in Recent Biosensor Research.

Citation	Method(s)	Setup	Parameters	Example/LOD	Equivalent Circuit/Notes
Magar et al. (2021) [40]	EIS	3-electrode	1 mHz–1 MHz; small sinusoidal	—	Rs, Rct, Cdl, Warburg
Brett (2022) [42]	EIS	Modified electrodes	100 kHz–10 mHz; 5–10 mV AC	—	RΩ, Rct, Cdl, Warburg, CPE
Lazanas & Prodromidis (2023) [43]	EIS	Various	10 μHz–1 MHz; 5–10 mV AC	Time constants 0.04 ms–0.4 s	Randles, Gerischer, K–K validation
Grieshaber et al. (2008) [44]	CV, CA, EIS	3-electrode	CV scan rate dependent; CA Cottrell response	LOD: 1.2 × 10^−7^ M (CV), 5.4 × 10^−7^ M (CA)	Randles; porous coatings
Randviir & Banks (2022) [45]	EIS	3-electrode	10–20 mV AC; 100 kHz–0.1 Hz	Rct equation relates conc.	Warns against oversimplification
Mocak et al. (1997) [46]	LOD/LOQ	—	k = 3 (LOD), k = 10 (LOQ)	Cd^2+^ detection via voltammetry	Justifies calibration/error bars
Chen et al. (2025) [47]	CV, EIS	Bio-integrated sensors	Real-time biomarker detection	DA: 1–106 nmol/L; Glu: 40–60 μM	Emphasis on antifouling, wireless
Bocu. (2023) [48]	Amperometry, EIS, FET	Bioelectrode interfaces	Enzyme/aptamer/CRISPR	±5% accuracy vs. ELISA	CRISPR, DET, apta-FETs

CV = Cyclic Voltammetry; CA = Chronoamperometry; EIS = Electrochemical Impedance Spectroscopy; FET = Field-Effect Transistor; Rs = Solution Resistance; Rct = Charge Transfer Resistance; Cdl = Double-Layer Capacitance; CPE = Constant Phase Element; LOD = Limit of Detection; LOQ = Limit of Quantification.

**Table 3 bioengineering-12-00988-t003:** Summarizes the primary mechanical, electrical, and procedural strategies explored in the literature for integrating biosensors into tibial components in TKA.

Focus Area	Sensor/System	Key Insight	TKA Relevance
Structural Integration	Embedded force sensors in tibial baseplate	Sensorized trays can be embedded without compromising joint geometry or load distribution; compatible with in vitro and cadaveric models.	Demonstrates safe embedding within tibial implants for force tracking.
Structural Integration	Piezoelectric cement interface detector	Smart groove sensor design enables monitoring of implant loosening without altering implant shape or footprint.	Supports use of minimal-form smart systems for failure detection.
Structural Integration	Parylene-C encapsulated electronics	Encapsulated sensors maintained function under >500,000 mechanical load cycles in simulated conditions.	Validates durability of packaging materials for long-term implant use.
Power & Data Transmission	Batteryless, lead-free biosensor designs	Optimal wireless implant design requires flexibility, SAR control, and stable packaging to avoid heat and mechanical strain.	Foundation for safe, wireless biosensor-powered TKA systems.
Power & Data Transmission	Wireless power transfer systems	Inductive and capacitive systems suitable for implantable sensors; trade-offs in range and tissue heating must be addressed.	Guides wireless powering strategies tailored to joint geometry.
Sterilization	Rotary implants post-autoclaving	Repeated sterilization degrades fatigue resistance and surface roughness; sensor packaging must resist heat and pressure.	Implies need for sterilization-compatible encapsulation.
Sterilization & Workflow	Printed Prussian Blue graphene sensors	Flexible, inkjet-printed sensors retained high conductivity and stability; potential for curved implant integration.	Supports adaptation of printed sensors to joint-conforming geometries.

**Table 4 bioengineering-12-00988-t004:** Preclinical and in vitro models of biosensor-enabled tibial implants.

Sensor Type	Model	Analyte/Data	Outcome	Limitation
pH, Temperature, Strain	Large animal (goat)	Physiological (mechanical + chemical)	Stable multi-sensor data transmission; no adverse mechanical effects	Short monitoring; small N
Lactate biosensor	Ex vivo bone	Lactate (infection marker)	Rapid infection detection; wireless	Ex vivo only
Nitric oxide biosensor	Rodent joint	Inflammatory response (NO)	Accurate detection; biodegradable confirmed	Single biomarker; rodent only
Triboelectric nanogenerator	Synthetic gait simulator	Load + energy harvesting	Sustained gait-based power; capacitive buffering	No biological validation
Piezoelectric biodegradable	Multi-source review (rodent & goat)	Mechanical strain; material degradation	Promising degradability, biocompatibility	No integrated system trials
Bioresorbable sensors	Rodent + review	Temporary NO and strain sensing	Biodegradable, non-immune	Short lifespan
General reviews	Multi-source	Miniaturization, wireless telemetry	Roadmap for implantable sensing	No new experimental validation

NO = nitric oxide; N = sample size.

**Table 5 bioengineering-12-00988-t005:** Human clinical evidence for sensor-integrated tibial implants.

Study	Device/Design	N/Setting	Primary Outcomes	Key Results	Take-Home
Wood et al. (2021) [67]	VERASENSE; RCT	~152	% balanced knees, PROMs	5.3% vs. 35.5% imbalance; PROMs same @12 mo	Balance rises; PROM parity
Afshari et al. (2022) [68]	VERASENSE; Multicentre RCT	250 (285 TKAs)	KOOS4, function	ΔKOOS4 = NS; 6MWT + 29 m; 9.4% vs. 36.8% imbalance	Balance rises; walk better; PROM parity
Sava et al. (2023) [69]	VERASENSE; Systematic review	14 studies; 3.6k+	PROMs, ROM, MUA, reop	PROMs improved both; MUA lower (1.4% vs. 5.0%)	PROM parity; MUA drops
Cho et al. (2018) [70]	VERASENSE; Prospective intraop	—	Ligament balance	Improved balance intraop; unclear long-term effect	Supports intraop precision
Cushner et al. (2024) [71]	Persona IQ; Case report	1	Recovery trajectory	Detected delayed recovery; enabled intervention	Proof-of-concept
Yocum et al. (2025) [72]	Persona IQ; Case series	3	KOOS-JR, gait, data	KOOS-JR > 90 @3 mo; 96–100% data; gait speed doubled	Telemetry feasible, promising
Guild et al. (2022) [73]	Persona IQ; Review	—	Design feasibility	FDA-cleared tibial telemetry; ~10 y battery; constraints	Establishes feasibility

KOOS4 = mean of KOOS pain, symptoms, ADL, and QoL subscales; PROMs = patient-reported outcome measures; ROM = range of motion; MUA = manipulation under anesthesia; 6MWT = six-minute walk test; FDA = U.S. Food and Drug Administration.

**Table 6 bioengineering-12-00988-t006:** Corrosion and Stability Testing Approaches for Implantable Sensor Materials.

Reference	Test(s)	Medium	Key Findings	Notes
ASTM F2129-19 [76].	Cyclic polarization.	Simulated electrolytes.	Identifies pitting and breakdown potentials.	Regulatory benchmark…
BioLogic (2024) [77].	Electrochemical Noise (EN, ZRA).	Acidic sulfate.	Noise resistance ~18 kΩ; ~3× lower than Rp.	Sensitive to early pit initiation…
Wackenrohr et al. (2022) [78].	OCP, EIS, corrosion-fatigue.	m-SBF.	Longer incubation, lower fatigue strength.	Defects dominate corrosion-fatigue…
Wackenrohr et al. (2024) [79].	Corrosion-fatigue with nanoparticles.	m-SBF.	CeO_2_ raises fatigue but lowers degradation; Fe_2_O_3_ raises fatigue.	Trade-offs from microstructural tailoring…
Talesh & Azadi (2024) [80].	Immersion + fatigue.	1× & 10× SBF.	PLA-coated Mg: +49% fatigue (no corrosion); −35% (corroded).	Accelerated immersion–fatigue tests critical…
Wang et al. (2025) [81].	Photopatternable PEDOT:PSS hydrogel.	Aqueous.	30 S/cm conductivity; 50% strain; stable.	Demonstrates electrochemical stability…
Li et al. (2025) [82].	Conductive hydrogel strategies.	—.	Hybrid hydrogels retain ~90% conductivity after 1000 cycles.	Requires fouling/hydrolysis checks…

OCP = Open Circuit Potential; EIS = Electrochemical Impedance Spectroscopy; EN = Electrochemical Noise; Rp = Polarization Resistance; SBF = Simulated Body Fluid.

**Table 7 bioengineering-12-00988-t007:** Biocompatibility and Hemocompatibility of Sensor Materials.

Citation.	Material/System.	Test(s).	Key Findings.	Compliance/Relevance…
ISO 10993-5:2009 [87].	General standard.	MEM elution, direct/indirect contact.	Defines ≥70% viability pass threshold; NRU/MTT/XTT assays standardized.	Regulatory benchmark…
Kohler et al. (2023) [53].	Parylene-C encapsulation.	ISO 10993-5 cytotoxicity.	Withstood 5 × 10^5^ cycles; non-cytotoxic; mechanically resilient.	Pass per ISO…
Li et al. (2025) [82].	PEDOT:PSS hydrogels (review).	Cytotoxicity, hemocompatibility.	Conductivity 4176 S/cm; tissue modulus compliance; hemocompatible.	Pass per ISO…
Menzel et al. (2024) [90].	Parylene-C on polyurethane.	ISO 10,993 cytotoxicity.	Suppressed toxic leachables; long-term CHO viability.	Pass per ISO…

MEM = Minimum Essential Medium; NRU = Neutral Red Uptake; MTT/XTT = Tetrazolium assays; CHO = Chinese hamster ovary cells; CNT = carbon nanotubes.

**Table 8 bioengineering-12-00988-t008:** Tribology and Tribocorrosion Testing Approaches in Implantable Components.

Citation.	Test(s).	Medium.	Key Findings.	Notes…
Takadoum (2023) [91].	Tribocorrosion review.	PBS/NaCl/Ringer ± proteins.	Protein adsorption can stabilize or accelerate corrosion; Nb/Zr/Ta alloys outperform Ti–6Al–4V.	Tribocorrosion linked to osseointegration loss…
Puthillam & Selvam (2024) [92].	Tribocorrosion mechanisms.	Various (albumin, H_2_O_2_, saline).	Adhesive, abrasive, fretting, cavitation wear interact; protein–peroxide accelerates Ti corrosion.	Highlights synergistic degradation…
Dreyer et al. (2024) [60].	In vivo vs. ISO wear simulation (5 M cycles).	Knee simulator vs. clinical dataset.	In vivo wear ~3× higher than ISO; anterior wear scar migration.	Shows ISO underestimates tribological stress…
Baykal et al. (2014) [93].	Pin-on-disk (multidirectional).	Serum vs. dry/water.	Cross-shear increases wear; highly crosslinked UHMWPE wear 0.67 vs. 5.39 mm^3^/MC.	Pin-on-disk bridges bench tests to simulator data…

MC = million cycles; UHMWPE = Ultra-High-Molecular-Weight Polyethylene; ISO = International Organization for Standardization; PBS = Phosphate-Buffered Saline.

**Table 9 bioengineering-12-00988-t009:** Antifouling and Stability Strategies for Implantable Electrochemical Biosensors.

Citation.	Material/Strategy.	Test/Environment.	Key Findings.	Relevance…
Yang et al. (2019) [95].	PDA + zwitterionic PSB coatings.	Protein adsorption assays, PBS, in vivo mouse implants.	Lowers protein adsorption (89%), Lowers fibroblast adhesion (86%); reduced inflammation; stable 4 weeks.	Zwitterionic > PEG for antifouling…
Jarosinska et al. (2024) [94].	PEG vs. sol–gel silicates, PLL-g-PEG.	CV, DPV, SWV in culture media.	PEG degraded in 72 h; silicate/PLL-g-PEG stable 6 weeks.	Long-term antifouling viable with silicate/PLL-g-PEG…
Fetter et al. (2024) [100].	Aptamer-based sensors.	Buffers with varied pH, ions, temp (33–41 °C).	Ionic/pH shifts negligible (<20% error); temperature drift dominant (63% error).	Temperature correction required…
Ye et al. (2024) [98].	Aptamer/CRISPR-DNA sensors.	Protein-rich media, in vivo microneedles.	Antifouling via DNA nanostructures; stable 14 days in vivo.	Nucleic acid antifouling strategies…
Liu et al. (2023) [99].	Electrode bonding + hybrid films.	Glutathione-rich solutions, protein media.	Au–C≡C & bidentate thiols: <6% drift; Au–S: ~23% drift.	Anchoring chemistries boost durability…
Campuzano et al. (2019) [96].	PEG hydrogels, zwitterionic peptides, SAMs.	Serum, blood, plasma assays.	PEDOT/PEG hydrogels maintained >90% current; zwitterionic peptides enabled fM detection; SAMs reduced nonspecific adsorption.	Multifunctional antifouling surfaces…

CV = cyclic voltammetry; DPV = differential pulse voltammetry; SWV = square-wave voltammetry; PDA = polydopamine; PSB = poly(sulfobetaine methacrylate); PEG = polyethylene glycol; PLL = poly-L-lysine; SAMs = self-assembled monolayers.

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
