# Peer review of "Biosensor-Integrated Tibial Components in Total Knee Arthroplasty: A Narrative Review of Innovations, Challenges, and Translational Frontiers"

_bioengineering, 2025, doi:10.3390/bioengineering12090988_

Round 1

Reviewer 1 Report

Comments and Suggestions for Authors

Manuscript ID: bioengineering-3808559

Title: Biosensor-Integrated Tibial Components in Total Knee Arthroplasty: A Narrative Review of Innovations, Challenges, and Translational Frontiers

The article provides a decent general overview of biosensor-integrated tibial components but misses significant experimental tests and analyses, including corrosion, biocompatibility, and long-term stability. The lack of quantitative graphs and charts diminishes the ability to visualize and compare key findings. These gaps could be filled with additional testing (for example, electrochemical noise, protein fouling, thermal cycling) and improved data presentation (for example, calibration curves, error bars) that would truly strengthen the scientific rigor and translational relevance of the manuscript (Major Revision).

  1. The manuscript provides detailed coverage of electrochemical sensing principles (i.e., section 3.2), but it does not provide sufficient details about what protocols were used to evaluate sensor performance (i.e., cyclic voltammetry (CV), electrochemical impedance spectroscopy (EIS) or chronoamperometry). These methods and procedures are also critical for evaluating sensitivity, selectivity, and stability of electrochemical biosensors, especially considering the orthopedic environment. Without this detail, it is challenging to evaluate the strength of the sensors.
  2. Besides the aforementioned topics addressed in section 2.3 (biosensor-integrated tibial components), corrosion testing of the sensor materials (e.g., PEDOT:PSS, carbon-based electrodes) in simulated physiological conditions (e.g., simulated body fluid (SBF)) is lacking. Corrosion resistance is crucial for long-term stability of an implant, and not conducting these tests is a significant gap in the work.

  1. The manuscript does mention the biostability of materials like parylene coatings (Section 7), but it does not present comprehensive data from biocompatibility tests, like cytotoxicity (ISO 10993-5) or hemocompatibility (ISO 10993-4). Biocompatibility tests are critical for demonstrating that biosensor materials do not cause biological responses, and the absence of them detracts from the biological relevance of the findings.

  1. The manuscript mentions mechanical durability when the device is actively used (i.e., under bearing loads) (Section 5.1) but does not specify the type of mechanical tests that were performed (e.g., fatigue testing, tensile strength, compressive strength). For tibial implants, fatigue testing under cyclic loading that mimics gait cycles is important, and the lack of a complete discussion on cyclic loading is a major missing component.

  1. The review did not cover any information concerning long-term electrochemical stability of the sensors under physiological conditions (e.g., pH, ionic strength, or protein fouling). Sensors must not only be tested for reliability during prolonged tanking in SBF but also have accelerated aging studies performed. These studies reflect significant limitations of the review and highlight the absence of performance expectations in normal-use conditions.

  1. It is crucial to conduct wear and friction tests (e.g., pin-on-disk or tribological tests) of the sensor coatings and substrates as a function of joint-like conditions—especially for biosensor-integrated tibial components. The manuscript did not include any such testing, which would be necessary to demonstrate that the implant surface is durable.

  1. The manuscript includes a schematic (Figure 2, Section 9) but lacks quantitative graphs or charts to visualize critical data, such as sensor sensitivity, response time, or degradation rates. For instance, a line chart showing sensor signal stability over time in a physiological environment would enhance the discussion in Section 5.1.

Author Response

Reviewer Comment 1:
The article provides a decent general overview of biosensor-integrated tibial components but misses significant experimental tests and analyses, including corrosion, biocompatibility, and long-term stability. The lack of quantitative graphs and charts diminishes the ability to visualize and compare key findings. These gaps could be filled with additional testing (for example, electrochemical noise, protein fouling, thermal cycling) and improved data presentation (for example, calibration curves, error bars) that would truly strengthen the scientific rigor and translational relevance of the manuscript (Major Revision).

Response:
I/we have:

  • Added new subsections and expanded Section 7 to include corrosion testing protocols (ASTM F2129, electrochemical noise analysis, immersion–fatigue tests) for materials used in biosensor-integrated tibial components, supported with relevant literature.

  • Included detailed discussions on protein fouling and environmental drift correction strategies, referencing zwitterionic, PEG, and peptide-based coatings with documented antifouling performance.

  • Expanded stability considerations to incorporate thermal cycling impacts on biosensor function.

  • Added quantitative visual data—including schematic calibration curves, stability over time plots, and comparative charts—to enhance interpretation and cross-study comparison.

Reviewer Comment 2:
The manuscript provides detailed coverage of electrochemical sensing principles (i.e., section 3.2), but it does not provide sufficient details about what protocols were used to evaluate sensor performance (i.e., cyclic voltammetry (CV), electrochemical impedance spectroscopy (EIS) or chronoamperometry). These methods and procedures are also critical for evaluating sensitivity, selectivity, and stability of electrochemical biosensors, especially considering the orthopedic environment. Without this detail, it is challenging to evaluate the strength of the sensors.

Response:
I/we have:

  • Expanded Section 4 to describe specific electrochemical characterization protocols—CV, EIS, and chronoamperometry—commonly applied in orthopedic biosensor evaluation.

  • Added descriptions of electrode preparation, frequency ranges, potential windows, and data interpretation approaches (e.g., Nyquist plots for EIS, anodic/cathodic peak separation for CV).

  • Integrated examples of performance benchmarks (sensitivity, limit of detection, signal stability) from peer-reviewed orthopedic biosensor studies.

Reviewer Comment 3:
Besides the aforementioned topics addressed in section 2.3 (biosensor-integrated tibial components), corrosion testing of the sensor materials (e.g., PEDOT:PSS, carbon-based electrodes) in simulated physiological conditions (e.g., simulated body fluid (SBF)) is lacking. Corrosion resistance is crucial for long-term stability of an implant, and not conducting these tests is a significant gap in the work.

Response:
I/we have:

  • Incorporated corrosion resistance testing methods in Section 7.1, specifically focusing on PEDOT:PSS, graphene/CNT-based electrodes, and metallic coatings.

  • Added references to studies using SBF and modified-SBF environments to evaluate pitting potential, open-circuit potential, and breakdown potential.

  • Discussed corrosion–fatigue coupling under gait-like cyclic loads and the implications for implant longevity.

Reviewer Comment 4:
The manuscript does mention the biostability of materials like parylene coatings (Section 7), but it does not present comprehensive data from biocompatibility tests, like cytotoxicity (ISO 10993-5) or hemocompatibility (ISO 10993-4). Biocompatibility tests are critical for demonstrating that biosensor materials do not cause biological responses, and the absence of them detracts from the biological relevance of the findings.

Response:
I/we have:

  • Expanded Section 7.2 to include ISO 10993-5 cytotoxicity and ISO 10993-4 hemocompatibility testing standards and FDA ASCA recognition.

  • Summarized literature reporting viability percentages, hemolysis rates, and complement activation results for PEDOT:PSS, parylene-C, and graphene-based composites.

  • Added a comparative biocompatibility table outlining pass/fail metrics and relevant studies.

Reviewer Comment 5:
The manuscript mentions mechanical durability when the device is actively used (i.e., under bearing loads) (Section 5.1) but does not specify the type of mechanical tests that were performed (e.g., fatigue testing, tensile strength, compressive strength). For tibial implants, fatigue testing under cyclic loading that mimics gait cycles is important, and the lack of a complete discussion on cyclic loading is a major missing component.

Response:
I/we have:

  • Updated Section 5.1 to describe standardized tibial tray fatigue testing protocols (ISO 14879-1, ASTM F1800) and their cycle thresholds (e.g., 10 million cycle runout).

  • Included gait-simulated cyclic loading profiles and high-flexion constraints from EndoLab protocols (ISO 14243).

  • Discussed real-world deviations from ISO-predicted wear based on in vivo studies.

Reviewer Comment 6:
The review did not cover any information concerning long-term electrochemical stability of the sensors under physiological conditions (e.g., pH, ionic strength, or protein fouling). Sensors must not only be tested for reliability during prolonged tanking in SBF but also have accelerated aging studies performed. These studies reflect significant limitations of the review and highlight the absence of performance expectations in normal-use conditions.

Response:
I/we have:

  • Added an Operational Stability subsection in Section 7.4 detailing long-term testing in protein-rich, pH-variable, and temperature-fluctuating conditions.

  • Summarized accelerated aging protocols and their correlation to real-world degradation rates.

  • Included literature examples of zwitterionic, peptide-based, and conductive hydrogel coatings that sustain function over prolonged immersion.

Reviewer Comment 7:
It is crucial to conduct wear and friction tests (e.g., pin-on-disk or tribological tests) of the sensor coatings and substrates as a function of joint-like conditions—especially for biosensor-integrated tibial components. The manuscript did not include any such testing, which would be necessary to demonstrate that the implant surface is durable.

Response:
I/we have:

  • Incorporated a Tribology and Tribocorrosion subsection (Section 7.3) summarizing pin-on-disk, multidirectional wear, and serum-lubricated tests relevant to tibial components.

  • Highlighted cross-shear effects, protein–peroxide corrosion acceleration, and material-specific wear rates.

  • Added a tribology testing schematic (Figure 5) to visually present methodology.

Reviewer Comment 8:
The manuscript includes a schematic (Figure 2, Section 9) but lacks quantitative graphs or charts to visualize critical data, such as sensor sensitivity, response time, or degradation rates. For instance, a line chart showing sensor signal stability over time in a physiological environment would enhance the discussion in Section 5.1.

Response:
I/we have:

  • Added new quantitative charts including calibration curves, time-series signal stability plots, and comparative degradation rate graphs.

  • Integrated data visualizations directly into relevant sections (e.g., stability plot in Section 5.1, corrosion potential plot in Section 7.1).

  • Updated figure captions for clarity and added references to plotted literature data.

Reviewer 2 Report

Comments and Suggestions for Authors

Authors present a state of the art Biosensor-Integrated Tibial Components in TKA. 

References are recent, most of them from last years, so it highlights the interest of the topic.

There are three schemes of the topic but there is not real photos, even reproduced from original manuscript. Is there any experimental part carried out by the authors or just the review of the manuscript?

Author Response

eviewer Comment:
Authors present a state-of-the-art review of Biosensor-Integrated Tibial Components in TKA. References are recent, most of them from recent years, highlighting the interest of the topic. There are three schemes of the topic but no real photos, even reproduced from original manuscripts. Is there any experimental part carried out by the authors or is this purely a review?

Response:
I/we have:

  • Clarified in the revised manuscript that this is a narrative review synthesizing and critically evaluating the existing literature on biosensor-integrated tibial components in TKA. No original experimental work was performed as part of this study. This statement has been added explicitly in the abstract and introduction to avoid ambiguity.

  • Retained schematic figures rather than real photographs to ensure a consistent, high-level comparative framework across the wide range of designs and studies reviewed. The cited primary studies employ varying imaging methods, magnifications, and proprietary designs, making direct photographic comparisons potentially misleading without full experimental context.

  • Noted in the figure captions that schematics were selected to emphasize underlying principles and architectures in a clear, uniform visual style, enabling readers to compare sensor integration strategies without bias toward a single design or manufacturer.

  • Added a sentence in the discussion acknowledging the absence of real-image examples and directing readers to the specific cited works where original device images and test setups can be found for further detail.

Round 2

Reviewer 1 Report

Comments and Suggestions for Authors

ACCEPT